# Development and comparison of single FLT3-inhibitors to dual FLT3/TAF1-inhibitors as an anti-leukemic approach

**Robert S. Leigh, Bogac L. Kaynak, Heikki Ruskoaho, Mika J. Välimäki**◉*

Drug Research Program, Division of Pharmacology and Pharmacotherapy, Faculty of Pharmacy, University of Helsinki, Helsinki, Finland

* mika.valimaki@helsinki.fi

## Abstract

Acute myeloid leukemia (AML) is characterized by several recurrent mutations that affect disease biology and phenotype, response to therapy and risk of subsequent relapse. Though tyrosine kinase inhibitors have gained regulatory approval for the treatment of AML, it is unclear whether single drugs targeting a specific genomic alteration will be sufficient to eradicate disease. Fortuitously, kinase/bromodomain inhibitors allow targeting of downstream transcriptional effectors of oncogenic pathways, allowing impediment of drug resistance at the transcriptional level. Successful development of combinatorial therapeutic strategies to inhibit both upstream oncogenic pathways and their downstream effectors could thus impede the onset of resistant disease. By using a combination of high-throughput cell-based screening assays and structure-based design, we have developed a novel anti-proliferative 3i-compound scaffold with a diverse range of single and dual FLT3/TAF1(2) activity against AML. Our novel approach to target both FLT3 kinase and TAF1(2) bromodomain efficiently maintained potency against haematological cancers. However, reference compounds and *in vitro* cell viability and cytotoxicity assays in cancer cell lines demonstrated superior effects of high affinity tyrosine kinase inhibition compared to inhibition of the TAF1 bromodomain. Our results highlight the feasibility of dual tyrosine kinase-bromodomain targeting to overcome disease mechanisms while also revealing the increased efficacy of FLT3-targeted compounds in AML.

## Introduction

Acute myeloid leukemia (AML) is a malignant disorder of hematopoietic stem cells characterized by clonal expansion of abnormally differentiated blasts of the myeloid lineage [1]. AML is generally treated using chemotherapeutic drugs (e.g., combination of cytarabine and an anthracycline). Unfortunately, with standard chemotherapy, long-term survival of patients with AML is achieved in only 35–45% of those younger than 60 years and 10–15% of those aged 60 years and older [2]. New drugs have recently been developed which act on genes necessary for AML cancer cell survival, such as FMS-like tyrosine kinase 3 (FLT3) and isocitrate dehydrogenase (IDH) 1/2 inhibitors [3–5]. However, the outcomes of patients with AML remain unsatisfactory due to the frequent occurrence of drug resistance mutations [6,7]

**Data availability statement:** All relevant data are within the manuscript and its Supporting Information files.

**Funding:** This work was supported by the Jane and Aatos Erkko Foundation (H.R, grant number -). The funder had no role in study design, data collection and analysis, decision to publish, or preparation of the manuscript. This work was supported by the Sigrid Juselius Foundation (H.R, grant number -). The funder had no role in study design, data collection and analysis, decision to publish, or preparation of the manuscript. This work was supported by the Finnish Cultural Foundation (R.S.L, grant number -). The funder had no role in study design, data collection and analysis, decision to publish, or preparation of the manuscript. This work was supported by the Finnish Foundation for Cardiovascular Research (R.S.L, grant number -). The funder had no role in study design, data collection and analysis, decision to publish, or preparation of the manuscript. There was no additional external funding received for this study.

**Competing interests:** The authors have declared that no competing interests exist.

with more than half of patients ultimately dying from their disease. Also, general toxicity of existing drugs prevents long-term treatment in many patients [1]. FLT3 belongs to a type III receptor tyrosine kinase (TK) family, whose members include structurally closely related c-KIT, FMS and platelet-derived growth factor (PDGF) receptors, enabling the comparison of protein templates and subsequent estimation of key structural features critical for affinity and selectivity. Type I FLT3 inhibitors, such as gilteritinib, target the active conformation of the FLT3 kinase domain and act against both internal tandem duplication (ITD) and tyrosine kinase domain (TKD) mutations. Type II FLT3 inhibitors, such as quizartinib, are specific for the inactive conformation and act only against ITD-mutations.

FLT3 activation, including in resistant forms, commonly exerts deleterious effects via activation of downstream transcriptional pathways mediated by signal transducer and activator of transcription (STAT5) and MYC [8–10]. Bromodomain inhibition allows for inhibition of MYC at the transcriptional level by impeding the interaction of bromodomain-containing proteins with acetylated lysine on histones, and this strategy has shown preclinical efficacy against several cancers [11–13]. Chemical targeting of bromodomains was demonstrated with the development of JQ1, an inhibitor of Bromodomain and extra terminal (BET) bromodomains [14], and BET Bromodomain inhibitors have been broadly reported to exhibit anti-cancer effects *in vitro* and *in vivo* [14,15]. Notably, BET bromodomain inhibition has been utilized in concert with TK inhibitors to overcome resistance-driving mutations in leukemia, and combinations of FLT3 and BET bromodomain inhibitors delivered as separate compounds are being explored as therapies [16–18].

In addition to BET bromodomain containing proteins (BRD2, BRD3 and BRD4), several dozen bromodomain-containing proteins exist in mammals, many of which have been targeted with chemical probes [19]. However, evidence for anti-oncogenic and combinatorial effects with TK inhibitors for most bromodomain inhibitors is lacking. TATA-binding protein associated factor 1 (TAF1) is a component of the TFIID transcriptional complex and contains a tandem bromodomain domain that has been successfully targeted with different scaffolds [20–26]. TAF1 is an essential gene for cell survival [27,28] and there are some reports that TAF1(2) bromodomain inhibition impedes oncogenesis with or without simultaneous inhibition of BET bromodomains [20,21,29,30]. There is also genetic evidence for the role of TAF1 in cancer, as mutations in TAF1 were implicated as drivers of clear cell endometrial cancer [31], and knockdown of TAF1 resulted in decreased proliferation and self-renewal in leukemia cells expressing a splice variant of the AML1-ETO fusion protein [32]. Additionally, TAF1 was shown to interact with acetylated p53 via its bromodomain to mediate transcriptional activation and promote p53 degradation via phosphorylation [33,34]. These studies imply that TAF1 inhibition might be a suitable therapeutic target in a broad set of cancers.

Chemical probes with dual target capability that can simultaneously inhibit both TKs and the TAF1(2) bromodomain would allow for the study of the added benefit of targeting the TAF1(2) bromodomain in cancer. We here utilize novel 3i-compounds with a diverse range of single and dual FLT3/TAF1 activity to investigate the potential anti-leukemic role of TAF1(2) bromodomain inhibition. Furthermore, we utilized reference compounds and *in vitro* cell viability and cytotoxicity assays in cancer cell lines, revealing that there is limited added benefit of TAF1(2) bromodomain inhibition in conjunction with high affinity TK inhibition.

## Materials and methods

### Computational chemistry

The commercial modelling package MOE 2022.02 (Chemical Computing Group Inc., Montreal, Canada) was utilized for protein modelling and docking experiments. The

docking protocol was carried out with a protein structure of FLT3 (PDB; 4XUF, type II) and TAF1(2) (PDB; 5I29) [35,36]. An Amber12:EHT forcefield was applied for the molecule parametrization and protein structure preparation. On-flight generated ligand conformations were placed in the cavity of the ATP binding site of FLT3 and the second bromodomain of TAF1(2) with the Triangle Matcher method and ranked with the London dG scoring function. Subsequently, the 30 highest ranked poses were applied for a refinement procedure containing the energy minimization and rescoring with the Generalized-Born Volume Integral/Weighted Surface area (GBVI/WSA dG) scoring function.

## Compound synthesis

Compound series of 3i-1244, 3i-1245, 3i-1246, 3i-1247 and 3i-1248 were synthesized by Enamine (Kyiv, Ukraine). Compound 3i-1103 was synthesized by the Faculty of Pharmacy, University of Helsinki [37] and reference compounds were obtained from commercial providers: GNE371 (Biosynth/Carbosynth), BAY299 (Tocris), Gilteritinib (ASP2215, Selleckchem) and Quizartinib (AC220, Selleckchem). All compounds conform to a minimum purity of >95% by suppliers.

## *In vitro* cancer cell efficacy assays

Drug sensitivity and resistance testing in MOLM13, DU4475, HDQP1, and IGROV1 cells was conducted at the Finnish Institute for Molecular Medicine (FIMM), University of Helsinki [38,39]. In brief, the effects of compounds on cell viability and cell death were quantified with Cell-Titer Glo (Promega) and CellTox Green (Promega), respectively. Compounds or vehicle control were included in cell culture medium at described concentrations for 72h using an automated liquid dispenser in 384-well format. For KASUMI-1 experiments, cells were grown in 96-well plates and similarly incubated with compounds at indicated concentrations for 72h in RPMI + 10% FCS + Gentamicin, followed by lysis and quantification of cell viability with Cell-Titer Glo (Promega).

## Generation of dose-response curves and data analysis

Raw values from viability and toxicity assays were normalized to DMSO and plotted to heat maps in R using pheatmap (RRID:SCR_016418). Dose-response curves were generated using the drc package in R [40]. In brief, nonlinear modelling was performed to calculate $IC_{50}$ values for each compound where possible and resulting curves from drc and experimental values were plotted using ggplot2 (RRID:SCR_014601).

## Kinase and bromodomain assays

The 3i-compounds underwent activity testing against a panel of 32 bromodomains using the Eurofins/DiscoverX bromoMAX assay, with specific focus on TAF1(2) bromodomain assessed by the KdELECT assay. In a similar manner, kinase activity was evaluated using a panel of 97 kinase proteins using the Eurofins/DiscoverX scanELECT and KdELECT assays, with specific focus on FLT3 protein. An 11-point 3-fold serial dilution of each test compound was prepared in 100% DMSO at 100x final test concentration and subsequently diluted to final concentration (DMSO 1%). $IC_{50}$ were determined using a compound top concentration 10,000 nM. Percent control was calculated using the following formula: ((test compound signal – positive control signal)/ (negative control signal – positive control signal)) $\times$ 100.

## Results

### FLT3/TAF1 bromodomain inhibitors with a novel chemical scaffold

Our aim was to produce a novel series of multitarget compounds for testing in AML assays and to provide insights on the molecular mechanisms of action and synergistic effects in treating multifactorial disease. We assumed that simultaneously targeting both FLT3 kinase and TAF1-bromodomains may potentially have a lower propensity to result in drug-resistant cancers. In our previous screening campaign, we identified a drug-like chemical probe, 3i-1103, as a modulator of atrial and ventricular reporter gene expression [37]. Preliminary structural analyses and ligand binding evaluations suggested a methoxyphenyl moiety of 3i-1103 as a structural feature (warhead) linked to TAF1 bromodomain and FLT3 kinase specificity (Fig 1). Subsequent docking experiments involving the compound 3i-1103 with the TAF1(2) bromodomain (PDB; 5I29) provided compelling evidence that the methoxyphenyl moiety could be successfully substituted with an acetyl lysine mimicking warhead [41,42]. This modification resulted in the most potent TAF1(2) interacting compound, 3i-1248, which establishes an anchoring interaction with Asn1604, exhibiting a calculated binding affinity of -6.73 kcal/mol (Fig 1A). Furthermore, docking studies of compound 3i-1103 with FLT3 kinase (PDB; 4XUF) identified a type II binding mode. The 3i-family compounds exhibited consistent docking scores against FLT3, with estimated binding affinities ranging from -7.60 kcal/mol to -8.24 kcal/mol (Fig 1A). It is noteworthy that compound 3i-1246 exhibited comparably strong calculated affinities towards both target proteins; however, the extensive protonation of this compound at physiological pH 7 compromised the reliability of these *in silico* results.

To further elucidate the compound's mechanism of action, we extended the chemical space around the hit candidate 3i-1103 by producing five novel 3i-compounds with various warhead moieties (*S1 Table*). Remarkably, the 3i-compound series displayed excellent selectivity among bromodomain- and kinase proteins [26]. We then investigated whether structural modifications to compound 3i-1103 could enhance its affinity for FLT3/TAF1(2) while preserving its selectivity among bromodomains and kinases. Comprehensive experimental testing revealed that compounds 3i-1244 and 3i-1245 demonstrated the lowest efficacy against FLT3 and TAF1(2), as detailed in *S1 Table*. In contrast, a methylpyridine-warhead carrying 4-(*tert*-butyl)-*N*-(3-(2-methylpyridin-4-yl)-1*H*-pyrazol-5-yl)benzamide (3i-1246) demonstrated a balanced FLT3/TAF1(2) activity profile, with $IC_{50}$ values of 18 nM and 280 nM, respectively. A phenyl-warhead carrying 4-(*tert*-butyl)-*N*-(3-phenyl-1*H*-pyrazol-5-yl)benzamide (3i-1247) showed a high affinity against FLT3 ($IC_{50}$ 1.5 nM), c-KIT ($IC_{50}$ 8.5 nM) and PDGFRB ($IC_{50}$ 3.4 nM). Finally, a 1-methyl-2-pyridone-warhead carrying 4-(tert-butyl)-N-(5-(1-methyl-6-oxo-1,6-dihydropyridin-3-yl)-1H-pyrazol-3-yl)benzamide (3i-1248) presented excellent affinity and selectivity against TAF1(2) ($IC_{50}$ 5.1 nM), while maintaining reasonable nM-level potency against FLT3 (Fig 1).

### Single and dual targeted inhibitors with distinct activity against tyrosine kinases and gain-of-function mutations

Due to structural consistency of the 3i-series and availability of FLT3/TAF1 protein structures, we were able to further confirm the ligand binding mode and predict the affinity through docking experiments (Fig 1). Our type II inhibitor 3i-1247 involved targeting the inactive form of the FLT3 kinase domain by utilizing both the ATP-binding site and adjacent pocket, demonstrating high selectivity comparable to other type II FLT3 inhibitors. However, other FLT3 type II-associated undesirable characteristics were also observed. Compound 3i-1247 was unable to overcome the insensitivity to TKD-mutations, and its selectivity for closely related c-KIT remained low. On the other hand, 3i-1248 with previously validated

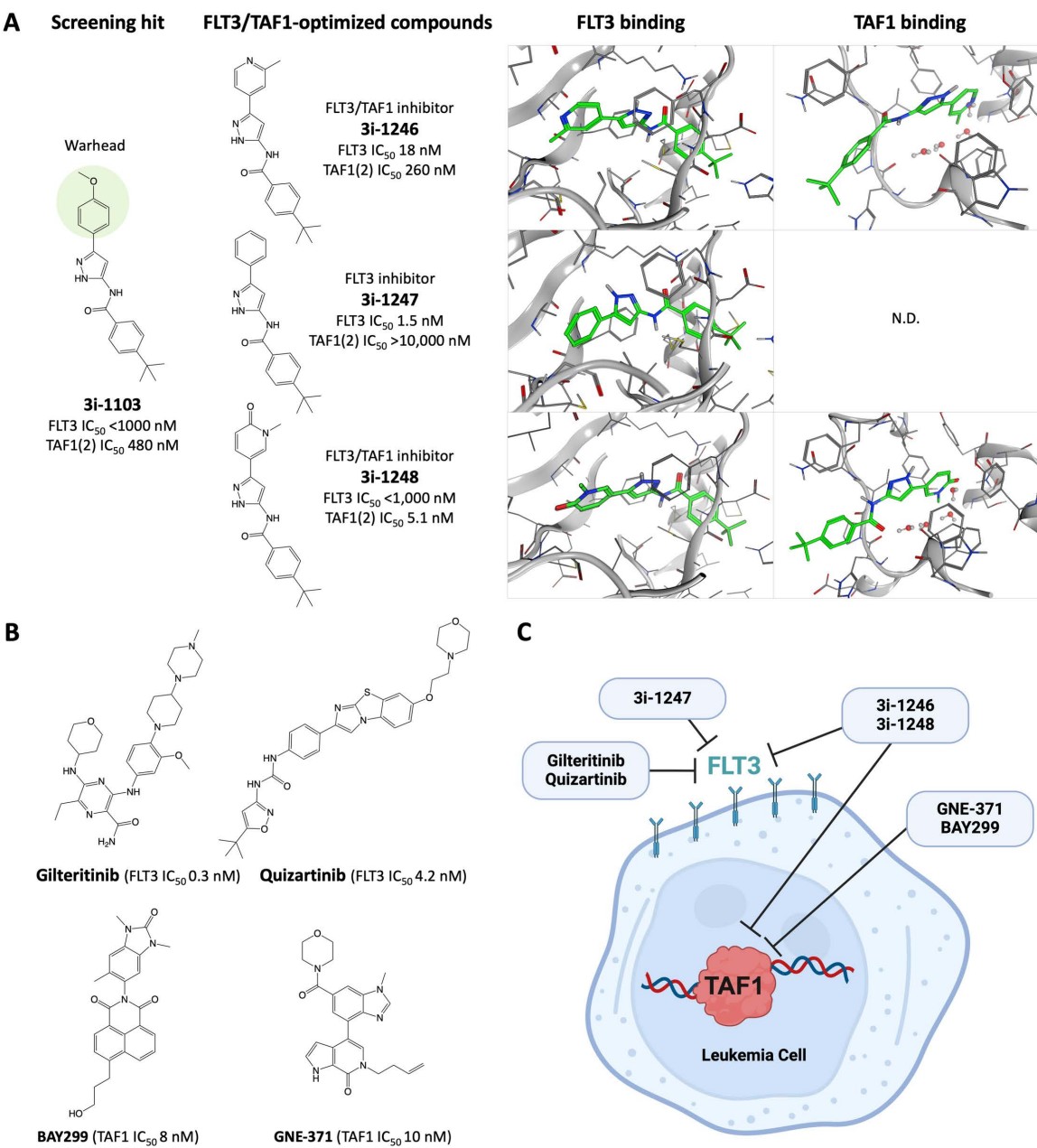

**Fig 1. Molecular structures and potency of novel single FLT3- and dual FLT3/TAF1(2)-inhibitors. (A)** A structurally focused chemical library was founded based on the previously identified screening hit 3i-1103 [37]. Novel 3i-compound series demonstrates an excellent half maximal inhibitory concentration (IC$_{50}$) and selectivity for FLT3- and TAF1(2)-inhibition. An 11-point, 3-fold serial dilution of each 3i-compound was assessed for IC$_{50}$ activity on FLT3 kinase and TAF1(2) bromodomain using assays conducted by Eurofins/DiscoverX in San Diego, USA. **(B)** Reference structures and literature-reported IC$_{50}$ values for gilteritinib (ASP2215), quizartinib (AC220), BAY299 and GNE-371 [20,24,43,44]. **(C)** A schematic representation of 3i-compounds exhibiting dual-target inhibition facilitates the investigation of the potential therapeutic benefits of targeting FLT3 kinase and TAF1(2) bromodomain in AML.

1-methyl-2-pyridone-warhead [21] dictates the TAF1(2) binding as expected. Therefore, the most potent compound in the cell-free assay, 3i-1248 (IC$_{50}$ 5.1 nM), remained our top-tier compound, providing high potency and selectivity towards the TAF1(2) bromodomain. As a result, we gathered a structurally consistent set of 3i-compounds with divergent target profiles.

These included compounds with moderate activity for FLT3 (3-1244, 3i-1245), a potent FLT3 inhibitor (3i-1247) and TAF1(2) targeted compounds with simultaneous FLT3 activity (3i-1103, 3i-1246, 3i-1248) (*S1 Table*). Activity against gain-of-function genetic mutations were also measured, revealing that novel compounds possessed diverse action against mutant FLT3 at 1 μM and 10 μM, including FLT3(835V), FLT3(ITD), FLT3(ITD, D835V), and FLT3(ITD, F691L) (Fig 2). We hypothesized that this set of 3i-compounds could reveal combinatorial effects of targeting FLT3 and TAF1(2) simultaneously in cancer cell assays.

## Effects of single tyrosine kinase and dual tyrosine kinase/bromodomain inhibition against AML

Single and dual TAF1/FLT3 inhibitors were tested for activity against MOLM13 leukemia cells based on reduction in Cell-titer Glo activity, which corresponds to the number of viable cells, and CellTox Green, which corresponds to the number of dead cells. Additionally, compounds were tested in KASUMI-1 cells, a leukemia cell line in which the TAF1(2) bromodomain had been previously implicated as a drug target [32]. In MOLM13 leukemia cells, FLT3 inhibitor 3i-1247 was the most potent compound overall, and possessed superb anti-proliferative activity compared to dual FLT3/TAF1 compounds 3i-1246 and 3i-1248 (Fig 3, *S1 Fig*). Concordantly, 3i-1246 (FLT3> TAF1) showed superior anti-cancer activity compared to the most potent TAF1(2) bromodomain inhibitor 3i-1248 (FLT3 < TAF1), suggesting the inability of TAF1(2) targeting to improve anti-proliferative effects in MOLM13 cells (Fig 3, *S1 Fig*).

## Efficacious anti-oncogenic activity of single and dual-targeted compounds is specific to leukemia cells

To assess whether TAF1(2) bromodomain inhibition shows an additional effect in non-AML lines, compounds were also tested in ovarian endometrioid adenocarcinoma (IGROV1) and breast carcinoma (HDQP1, DU4475) cell lines. Compounds displayed only mild activity in these cell lines, though on a relative basis single FLT3-targeted compound 3i-1247 showed superior activity in DU4475 cells compared to dual FLT3/TAF1(2) inhibitors (3i-1103, 3i-1246, 3i-1248) (Fig 4, *S2 Fig*). Curiously, dual FLT3/TAF1(2) inhibitors 3i-1246 and 3i-1248 were more active than FLT3 inhibitor 3i-1247 in HDQP1 cells, though their IC$_{50}$ was >10 μM, precluding therapeutic utility but potentially indicating anti-proliferative effects of bromodomain inhibition in HDQP1 cells (*S3 and S4 Figs*). All compounds showed similarly low activity in IGROV1 cells (*S5 and S6 Figs*).

## Superior activity of single FLT3 targeted compounds against KASUMI-1 cells possessing the AML1-ETO fusion transcription factor

The development of targeted therapies for the treatment of specific cancer subtypes based on their genetic background represents an important therapeutic strategy. TAF1 was recently observed to be a key target in leukemia cells possessing the AML1-ETO fusion transcription factor by modulation of protein-protein interactions [32]. Based on this finding, experiments were conducted in KASUMI-1 cells, which contain the AML1-ETO transcription factor (*S7 Fig*). Our hypothesis was that dual FLT3/TAF1(2) inhibitors would more potently inhibit KASUMI-1 growth due to dual activity against both TKs and the TAF1 bromodomain by inhibiting interaction between AML1-ETO and the TAF1 bromodomain. However, the most potent FLT3-targeted compound 3i-1247 showed superior anti-proliferative activity to dual FLT3/TAF1(2) compounds 3i-1103, 3i-1246, and 3i-1248 in KASUMI-1 cells (*S7 Fig*), similar to results in MOLM13, DU4475, and IGROV1 cell lines (Figs 3 and 4, *S1-S6 Figs*). Collectively,

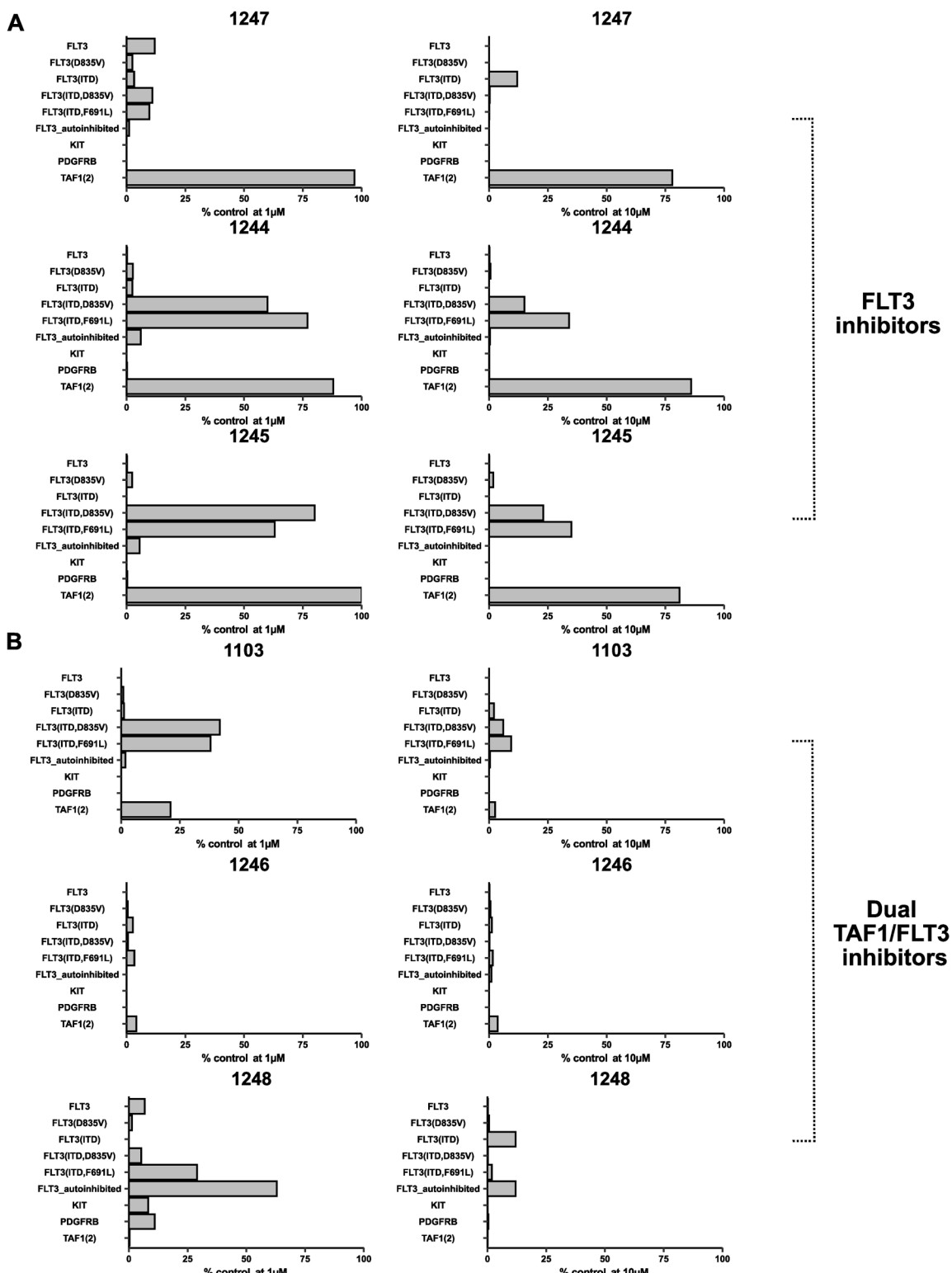

**Fig 2. Results of 3i-compound series at concentrations of 1 μM and 10 μM in a cell free Eurofins/DiscoverX assay for kinase and bromodomain proteins. (A)** Single-targeted compounds with tyrosine kinase activity but without bromodomain activity. **(B)** Dual-targeted compounds with both tyrosine kinase and bromodomain activity.

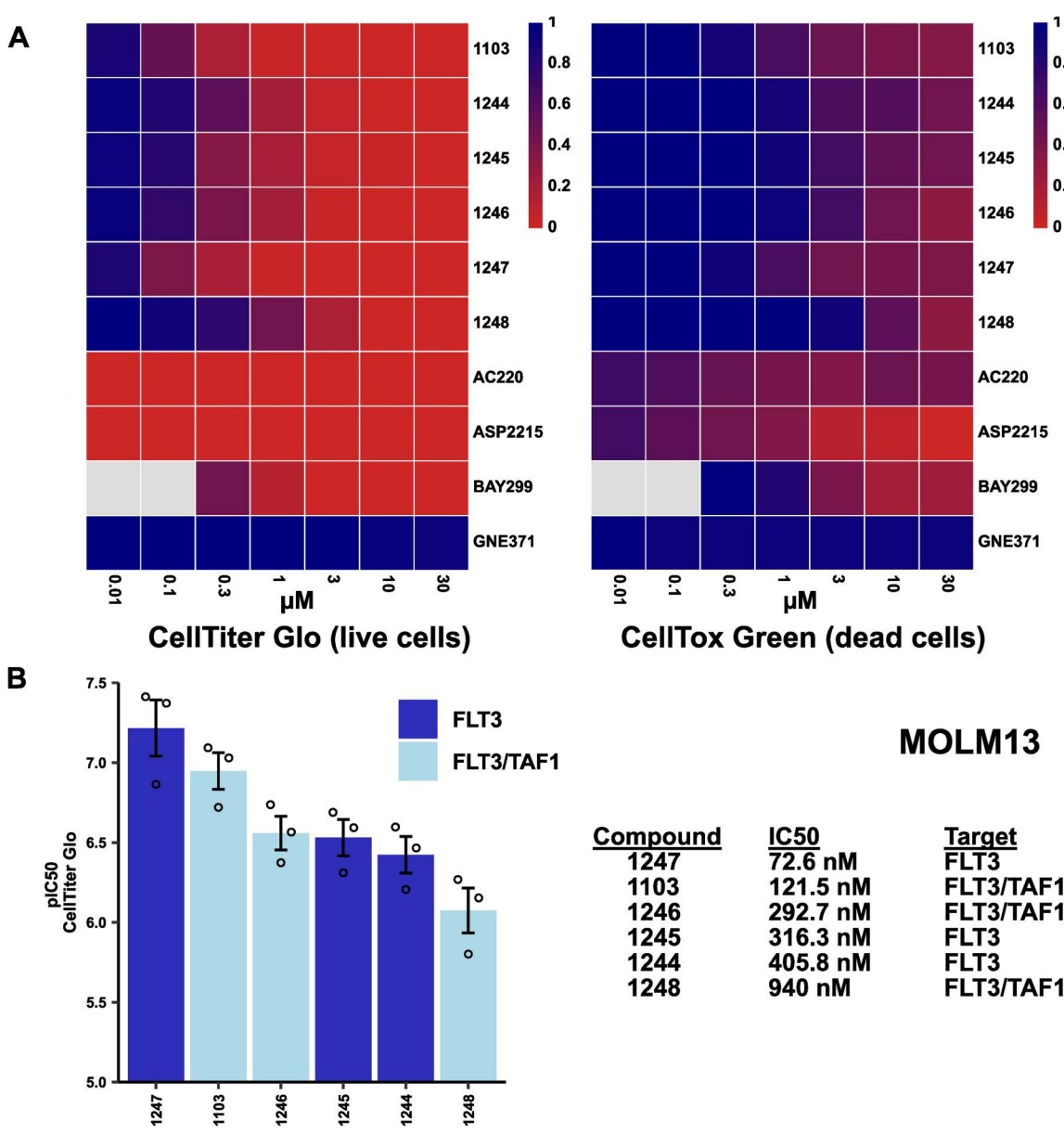

**Fig 3. Anti-leukemic activity of novel compounds and reference drugs *in vitro*.** MOLM13 cells were treated with a series of FLT3/TAF1 compounds and cell viability was determined using a CellTiter Glo assay, while cell toxicity was determined using a CellTox Green assay. **(A)** Compound activity in MOLM13 cells at 0.01-30 μM (n ≥ 3). **(B)** $pIC_{50}$ and $IC_{50}$ values of FLT3/TAF1 inhibitors. Dots represent independent experiments (n = 3) and mean ± SEM is shown. Both single FLT3 inhibitors (3i-1244, 3i-1245, 3i-1247) and dual FLT3/TAF1 inhibitors (3i-1103, 3i-1246, 3i-1248) were used for experiments. Tyrosine kinase activity (3i-1247, AC220, ASP2215) rather than TAF1(2) bromodomain inhibition (3i-1248, GNE371) was determined as the most relevant protein target in this assay.

these results show little added benefit to targeting the TAF1(2) bromodomain in addition to FLT3 in a single compound to achieve anti-oncogenic effects.

## Discussion

By using a combination of high-throughput cell-based screening assays and structure-based design, we have uncovered a novel molecular scaffold against AML. At low concentrations,

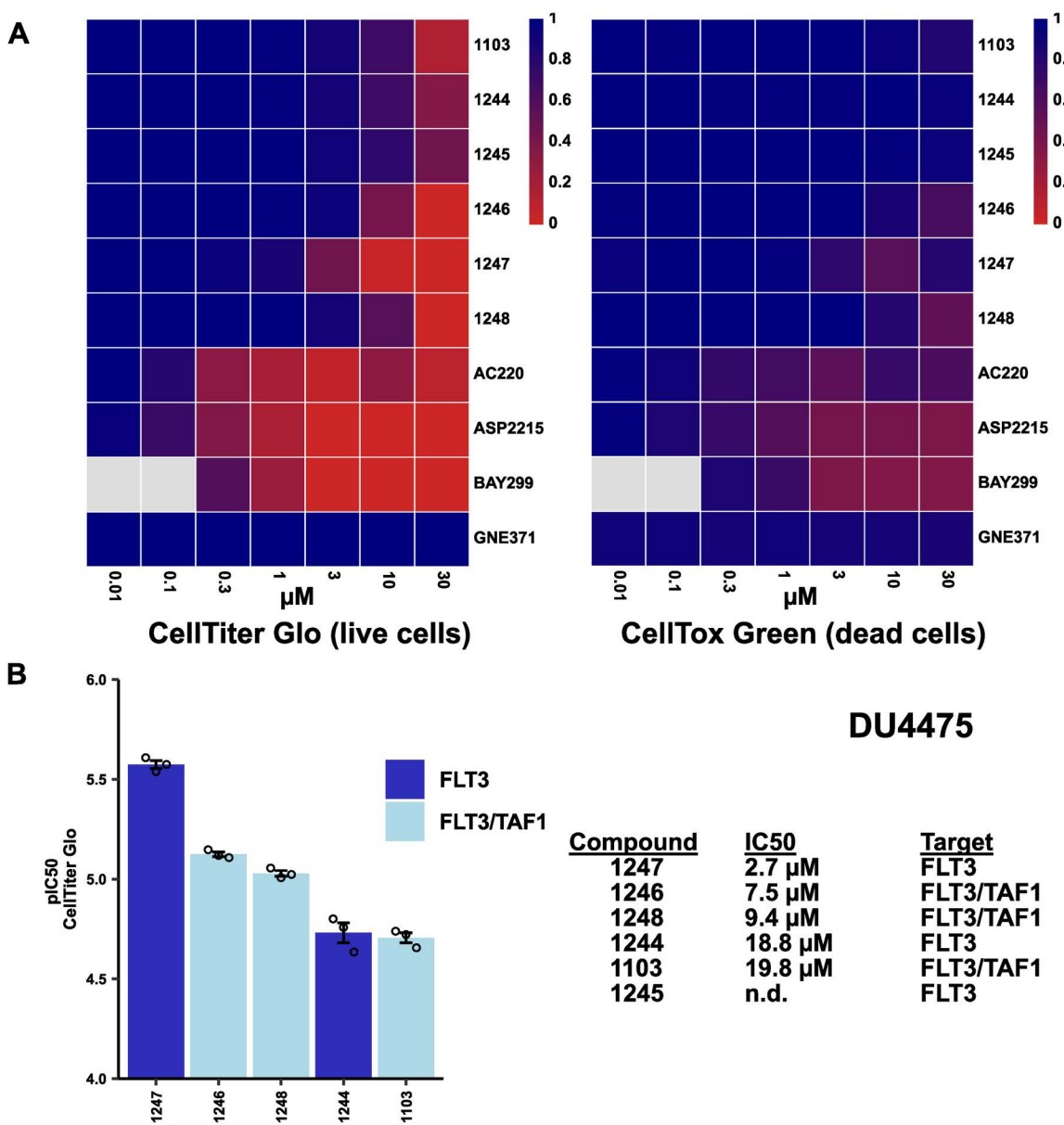

**Fig 4. Anti-oncogenic activity of novel compounds and reference drugs *in vitro*. DU4475 cells were treated with a series of FLT3/TAF1 compounds and cell viability was determined using a CellTiter Glo assay, while cell toxicity was determined using a CellTox Green assay. (A)** Compound activity in DU4475 cells at 0.01-30 μM (n ≥ 3). **(B)** $pIC_{50}$ and $IC_{50}$ values of FLT3/TAF1 inhibitors. Dots represent independent experiments (n = 3) and mean ± SEM is shown. Both single FLT3 inhibitors (3i-1244, 3i-1245, 3i-1247) and dual FLT3/TAF1 inhibitors (3i-1103, 3i-1246, 3i-1248) were used for experiments. Compounds for which $IC_{50}$ values could not be generated for three experiments or which were determined to be greater than the maximal tested concentration (30 μM) are listed as n.d. (not determined). Tyrosine kinase activity (3i-1247, AC220, ASP2215) rather than TAF1(2) bromodomain inhibition (3i-1248, GNE371) was determined as the most relevant protein target in this assay.

these 3i-compounds efficiently inhibit the growth of AML cell lines. Interestingly, bromodomain and kinase screening results indicated that 3i-compounds act on a set of clinically relevant AML drug targets: FLT3, TAF1, c-KIT and PDGFRB. By novelly targeting both FLT3 kinase and the TAF1 bromodomain, we anticipated that these novel agents may have a lower propensity to result in drug-resistant cancers. Additionally, using a pluripotent stem

cell-based toxicological screening assay, these novel 3i-compounds are less toxic to stem cells than tool compound BAY299 [26].

Taken together, our results indicated that TAF1(2) bromodomain inhibition does not have anti-proliferative effects in tested cell lines. This is in contrast to the TAF1 protein itself, which is an essential gene that is required for cell cycle progression [27,28]. However, our additional data suggest the TAF1 bromodomain functions in the regulation of developmental gene expression [26], and reporter gene assays conducted with TAF1 cDNA indicate that TAF1 also functions as a transcriptional repressor. Inhibition of the bromodomain (by chemical compounds or mutation) impedes these repressive activities, resulting in epigenetic upregulation of gene expression. Additionally, our results indicate that to maximize epigenetic effects the mutation of both TAF1 bromodomains is necessary. It is conceivable that by inactivating both TAF1 bromodomains, cells could undergo apoptosis by epigenetic upregulation of, e.g., tumor suppressor genes. However, no available assays exist that would allow optimization of our chemical scaffold for the first TAF1 bromodomain, and due to the druggability issue of the first TAF1 bromodomain [25], it is unlikely that this would be feasible in a therapeutic setting. Additionally, it is possible that even following inhibition of both bromodomains, other functions of TAF1 are sufficient to maintain cancer cell proliferation. This is supported by a recent report which indicates that even when the bromodomain of BRD4 is inhibited, its interaction with transcriptional co-activators in breast cancer is unimpeded due to the presence of a longer isoform capable of maintaining protein-protein interactions [45]. Our results suggest that TAF1 function, essential for cell survival, also remains intact in the presence of bromodomain inhibition. Admittedly, only a few cell lines were tested in the present study, and it is conceivable that dual FLT3/TAF1 inhibitors might display improved activity over single FLT3 inhibitors at therapeutically relevant concentrations in cell lines with specific genetic mutations. Additionally, different combinations of TK/bromodomain inhibitors might more effectively induce anti-proliferative effects. Further insight into the requirement of specific bromodomain-containing proteins and the bromodomain itself in cancer subtypes would greatly enhance efforts to develop new therapies targeting the transcriptional machinery. However, based on the results of the present study, we concluded that added benefit of TAF1(2) bromodomain inhibition is minimal for the development of anti-leukemic compounds and further focused on optimizing compound activity against tyrosine kinases.

## Supporting information

**S1 Table. Molecular structures, tyrosine kinase- and bromodomain inhibition of compounds synthesized by University of Helsinki and Enamine (Kyiv, Ukraine).** Inhibition constant ($IC_{50}$) of novel 3i-compounds against FLT3 and TAF1(2) (kdELECT, Eurofins/DiscoverX, USA). (PDF)

**S1 Fig. Anti-oncogenic activity of novel compounds and reference drugs *in vitro*. MOLM13 cells were treated with a series of FLT3/TAF1 compounds and cell viability was determined using a CellTiter Glo assay.** Dose response curves were generated in triplicate, and each dose-response curve represents an independent experiment. Dots represent individual data points and the gray area represents the 95% confidence interval for each curve. Both single FLT3 inhibitors (3i-1244, 3i-1245, 3i-1247) and dual FLT3/TAF1 inhibitors (3i-1103, 3i-1246, 3i-1248) were used for experiments. (PDF)

**S2 Fig. Anti-oncogenic activity of novel compounds and reference drugs *in vitro*. DU4475 cells were treated with a series of FLT3/TAF1 compounds and cell viability was determined**

using a CellTiter Glo assay. Dose response curves were generated in triplicate, and each dose-response curve represents an independent experiment. Dots represent individual data points and the gray area represents the 95% confidence interval for each curve. Both single FLT3 inhibitors (3i-1244, 3i-1245, 3i-1247) and dual FLT3/TAF1 inhibitors (3i-1103, 3i-1246, 3i-1248) were used for experiments.
(PDF)

**S3 Fig. Anti-oncogenic activity of novel compounds and reference drugs *in vitro*. HDQP1 cells were treated with a series of FLT3/TAF1 compounds and cell viability was determined using a Cell-titer Glo assay, while cell toxicity was determined using a CellTox Green assay.** (**A**) Compound activity in HDQP1 cells at 0.01-30 μM (n ≥ 3). (**B**) $pIC_{50}$ and $IC_{50}$ values of FLT3/TAF1 inhibitors. Dots represent independent experiments (n = 3) and mean ± SEM is shown. Both single FLT3 inhibitors (3i-1244, 3i-1245, 3i-1247) and dual FLT3/TAF1 inhibitors (3i-1103, 3i-1246, 3i-1248) were used for experiments. Compounds for which $IC_{50}$ values could not be generated for three experiments or which were determined to be greater than the maximal tested concentration (30 μM) are listed as n.d. (not determined).
(PDF)

**S4 Fig. Anti-oncogenic activity of novel compounds and reference drugs *in vitro*. HDQP1 cells were treated with a series of FLT3/TAF1 compounds and cell viability was determined using a CellTiter Glo assay.** Dose response curves were generated in triplicate, and each dose-response curve represents an independent experiment. Dots represent individual data points and the gray area represents the 95% confidence interval for each curve. Both single FLT3 inhibitors (3i-1244, 3i-1245, 3i-1247) and dual FLT3/TAF1 inhibitors (3i-1103, 3i-1246, 3i-1248) were used for experiments. Compounds for which $IC_{50}$ values could not be generated are listed as n.d. (not determined).
(PDF)

**S5 Fig. Anti-oncogenic activity of novel compounds and reference drugs *in vitro*. IGROV1 cells were treated with a series of FLT3/TAF1 compounds and cell viability was determined using a Cell-titer Glo assay, while cell toxicity was determined using a CellTox Green assay.** (**A**) Compound activity in IGROV1 cells at 0.01-30 μM (n ≥ 3). (**B**) $pIC_{50}$ and $IC_{50}$ values of FLT3/TAF1 inhibitors. Dots represent independent experiments (n = 3) and mean ± SEM is shown. Both single FLT3 inhibitors (3i-1244, 3i-1245, 3i-1247) and dual FLT3/TAF1 inhibitors (3i-1103, 3i-1246, 3i-1248) were used for experiments. Compounds for which $IC_{50}$ values could not be generated for three experiments or which were determined to be greater than the maximal tested concentration (30 μM) are listed as n.d. (not determined).
(PDF)

**S6 Fig. Anti-oncogenic activity of novel compounds and reference drugs *in vitro*. IGROV1 cells were treated with a series of FLT3/TAF1 compounds and cell viability was determined using a CellTiter Glo assay.** Dose response curves were generated in triplicate, and each dose-response curve represents an independent experiment. Dots represent individual data points, and the gray area represents the 95% confidence interval for each curve. Both single FLT3 inhibitors (3i-1244, 3i-1245, 3i-1247) and dual FLT3/TAF1 inhibitors (3i-1103, 3i-1246, 3i-1248) were used for experiments.
(PDF)

**S7 Fig. Anti-oncogenic activity of novel compounds and reference drugs *in vitro*. KASUMI-1 cells were treated with a series of FLT3/TAF1 compounds and cell viability was determined using a CellTiter Glo assay, while cell toxicity was determined using a CellTox**

**Green assay.** (**A**) Compound activity in KASUMI-1 cells at 0.001-30 μM (n = 1). (**B**) $IC_{50}$ values of FLT3/TAF1 inhibitors. Each dose-response curve represents an independent experiment. Dots represent individual data points and the gray area represents the 95% confidence interval for each curve. Both single FLT3 inhibitors (3i-1244, 3i-1245, 3i-1247) and dual FLT3/TAF1 inhibitors (3i-1103, 3i-1246, 3i-1248) were used for experiments.
(PDF)

**S1 Appendix. Individual values for figures.**
(XLSX)

## Acknowledgments

We are grateful to the High Throughput Biomedicine Unit at the Finnish Institute for Molecular Medicine (University of Helsinki) for assistance with *in vitro* cancer cell efficacy assays.

## Author contributions

**Conceptualization:** Robert S. Leigh, Bogac L. Kaynak, Heikki Ruskoaho, Mika J. Välimäki.

**Data curation:** Robert S. Leigh, Mika J. Välimäki.

**Formal analysis:** Robert S. Leigh, Bogac L. Kaynak, Heikki Ruskoaho, Mika J. Välimäki.

**Funding acquisition:** Robert S. Leigh, Bogac L. Kaynak, Heikki Ruskoaho.

**Investigation:** Robert S. Leigh, Mika J. Välimäki.

**Writing – original draft:** Robert S. Leigh, Mika J. Välimäki.

**Writing – review & editing:** Robert S. Leigh, Bogac L. Kaynak, Heikki Ruskoaho, Mika J. Välimäki.

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
