## [Decision Letter · Decision Letter 0]

30 Dec 2024

PONE-D-24-53115Development and comparison of single FLT3-inhibitors to dual FLT3/TAF1 inhibitors as an anti-leukemic approachPLOS ONE

Dear Dr. Välimäki,

Thank you for submitting your manuscript to PLOS ONE. After careful consideration, we feel that it has merit but does not fully meet PLOS ONE’s publication criteria as it currently stands. Therefore, we invite you to submit a revised version of the manuscript that addresses the points raised during the review process by the two Reviewers, experts in the field.

We look forward to receiving your revised manuscript.

Kind regards,

Francesco Bertolini, MD, PhD

Academic Editor

PLOS ONE

Journal Requirements:

"This work was supported by the Jane and Aatos Erkko Foundation, and the Sigrid Juselius Foundation (HR). R.S.L was supported by personal grants from the Finnish Cultural Foundation and the Finnish Foundation for Cardiovascular Research. "

"This work was supported by the Jane and Aatos Erkko Foundation, and the Sigrid Juselius Foundation. R.S.L was supported by personal grants from the Finnish Cultural Foundation and the Finnish Foundation for Cardiovascular Research. We are grateful to the High Throughput Biomedicine Unit at the Finnish Institute for Molecular Medicine (University of Helsinki) for assistance with in vitro cancer cell efficacy assays. "

"This work was supported by the Jane and Aatos Erkko Foundation, and the Sigrid Juselius Foundation (HR). R.S.L was supported by personal grants from the Finnish Cultural Foundation and the Finnish Foundation for Cardiovascular Research. "

5. We note that your Data Availability Statement is currently as follows: All relevant data are within the manuscript and its Supporting Information files.

6. We notice that your supplementary figures are uploaded with the file type 'Figure'. Please amend the file type to 'Supporting Information'. Please ensure that each Supporting Information file has a legend listed in the manuscript after the references list.

Reviewers' comments:

Reviewer's Responses to Questions

**Comments to the Author**

1. Is the manuscript technically sound, and do the data support the conclusions?

Reviewer #1: Yes

Reviewer #2: Partly

2. Has the statistical analysis been performed appropriately and rigorously? 

Reviewer #1: I Don't Know

Reviewer #2: No

3. Have the authors made all data underlying the findings in their manuscript fully available?

Reviewer #1: Yes

Reviewer #2: Yes

4. Is the manuscript presented in an intelligible fashion and written in standard English?

Reviewer #1: Yes

Reviewer #2: Yes

5. Review Comments to the Author

Reviewer #1: This paper nicely describes the discovery of compounds that are dual FLT3/TAF1 (bromodomain) inhibitors for certain leukemia cell lines and the comparison of the single-mechanism inhibitors. The manuscript is very brief but well written and understandable. I recommend publication but not until a few specific issues are resolved.

1) There needs to be a much better description of what bromodomains are (BET abbreviation defined), what they do and how/why their inhibitors are used for hematological cancers.

2) Related to (1), a discussion of why the authors thought it may be useful to target both proteins (TK and BETs) other than a report that it may reduce resistance, is needed. The paper would benefit greatly from a figure that shows both pathways and their possible link(?) in a synergistic effect in anti-leukemia therapeutic design.

3) The authors do not discuss the computational chemistry at all, save for a sentence stating that the methoxyphenyl group looks like he "warhead". The authors need to tie this in with previous work and show why these structures are relevant by having a paragraph dedicate to the modeling

4) At least one of the figures with heat maps and inhibition curves can be put into the supplemental information. Some graphs do not have calculated IC50's---please state why

5) As far as I can see in the manuscript, No statistical analysis was performed on activity of the compounds...were the curves generated once or in duplicate, triplicate?

The paper shows negative results, but these are still relevant. So I recommend publication, but much more details are needed to put this entire concept the authors posit into a proper therapeutic context.

Reviewer #2: The authors did in vitro analysis of single and dual FLT3/TAF1 inhibitors for acute myeloid leukemia (AML). They tested several 3i-compounds and provide data on their inhibitory effects on FLT3 kinase and TAF1 bromodomains using cell lines. They also found that there is limited added benefit of TAF1(2) bromodomain inhibition in conjunction with high affinity TK inhibition.

Below are some major comments:

- The manuscript concludes that TAF1 bromodomain inhibition does not significantly enhance anti-leukemic effects when combined with FLT3 inhibition. However, it could be more thoroughly discussed, especially across different cell lines, and from other literature that might include more mechanistic analysis on the TAF1 inhibition? Becaue there is lack of mechanistic studies in this paper, I wonder if authors may have any hypothesis of genetic or proteomic explanations why TAF1 targeting offers minimal benefit?

- The main conclusion were mainly from cell lines such as MOLM13 and KASUMI-1, and I'm not sure if that is sufficient to draw such conclusions, might need to discuss that, especially if further analysis (with primary cells/in vivo) will be needed to confirm the finding.

- The statistical analysis is unclear, and also the sample size, confidence intervals or p-values for the IC50 differences should be included to substantiate claims.

- In fig. 1 legends, could author also describe how they obtain the IC50 numbers?

6. PLOS authors have the option to publish the peer review history of their article (what does this mean? ). If published, this will include your full peer review and any attached files.

**Do you want your identity to be public for this peer review?** For information about this choice, including consent withdrawal, please see our Privacy Policy .

Reviewer #1: **Yes: ** Joe Barchi

Reviewer #2: No

---

## [Author Response · Author response to Decision Letter 1]

11 Feb 2025

Manuscript ID: PONE-D-24-53115

Title: "Development and comparison of single FLT3-inhibitors to dual FLT3/TAF1 inhibitors as an anti-leukemic approach"

Reviewer: 1

Comments:

This paper nicely describes the discovery of compounds that are dual FLT3/TAF1 (bromodomain) inhibitors for certain leukemia cell lines and the comparison of the single-mechanism inhibitors. The manuscript is very brief but well written and understandable. I recommend publication but not until a few specific issues are resolved.

Specific issues:

1. There needs to be a much better description of what bromodomains are (BET abbreviation defined), what they do and how/why their inhibitors are used for hematological cancers.

Response: We agree that this could be better described in the introduction to the manuscript and have added additional text and literature references on the mechanistic basis for bromodomain inhibition in hematological and other cancers. This information has been added into the revised manuscript on lines 52-63.

“FLT3 activation, including in resistant forms, commonly exerts deleterious effects via activation of downstream transcriptional pathways mediated by signal transducer and activator of transcription (STAT5) and MYC [8–10]. Bromodomain inhibition allows for inhibition of MYC at the transcriptional level by impeding the interaction of bromodomain-containing proteins with acetylated lysine on histones, and this strategy has shown preclinical efficacy against several cancers [11–13]. Chemical targeting of bromodomains was demonstrated with the development of JQ1, an inhibitor of Bromodomain and extra terminal (BET) bromodomains [14], and BET Bromodomain inhibitors have been broadly reported to exhibit anti-cancer effects in vitro and in vivo [14,15]. Notably, BET bromodomain inhibition has been utilized in concert with TK inhibitors to overcome resistance-driving mutations in leukemia, and combinations of FLT3 and BET bromodomain inhibitors delivered as separate compounds are being explored as therapies [16–18].”

2. Related to (1), a discussion of why the authors thought it may be useful to target both proteins (TK and BETs) other than a report that it may reduce resistance, is needed. The paper would benefit greatly from a figure that shows both pathways and their possible link(?) in a synergistic effect in anti-leukemia therapeutic design.

Response: We have included a discussion of this point in the additional text (lines 368-374).

“Additionally, it is possible that even following inhibition of both bromodomains, other functions of TAF1 are sufficient to maintain cancer cell proliferation. This is supported by a recent report which indicates that even when the bromodomain of BRD4 is inhibited, its interaction with transcriptional co-activators in breast cancer is unimpeded due to the presence of a longer isoform capable of maintaining protein-protein interactions [45]. Our results suggest that TAF1 function, essential for cell survival, also remains intact in the presence of bromodomain inhibition.”

Furthermore, Figure 1C along with corresponding figure legend have been added into the manuscript (lines 205-209). “A schematic representation of 3i-compounds exhibiting dual-target inhibition facilitates the investigation of the potential therapeutic benefits of targeting FLT3 kinase and TAF1(2) bromodomain in AML.

3. The authors do not discuss the computational chemistry at all, save for a sentence stating that the methoxyphenyl group looks like the "warhead". The authors need to tie this in with previous work and show why these structures are relevant by having a paragraph dedicate to the modeling.

Response: Thank you for pointing this out. As requested by the Reviewer, we have now incorporated the key findings pertaining to the docking experiments into the revised manuscript (lines 166-176) as follows:

“Preliminary structural analyses and ligand binding evaluations suggested a methoxyphenyl moiety of 3i-1103 as a structural feature (warhead) linked to TAF1 bromodomain and FLT3 kinase specificity (Fig 1). Subsequent docking experiments involving the compound 3i-1103 with the TAF1(2) bromodomain (PDB; 5I29) provided compelling evidence that the methoxyphenyl moiety could be successfully substituted with an acetyl lysine mimicking warhead [41,42]. This modification resulted in the most potent TAF1(2) interacting compound, 3i-1248, which establishes an anchoring interaction with Asn1604, exhibiting a calculated binding affinity of -6.73 kcal/mol (Fig. 1A). Furthermore, docking studies of compound 3i-1103 with FLT3 kinase (PDB; 4XUF) identified a type II binding mode. The 3i-family compounds exhibited consistent docking scores against FLT3, with estimated binding affinities ranging from -7.60 kcal/mol to -8.24 kcal/mol (Fig. 1A). It is noteworthy that compound 3i-1246 exhibited comparably strong calculated affinities towards both target proteins; however, the extensive protonation of this compound at physiological pH 7 compromised the reliability of these in silico results.”

To further clarify kinase and bromodomain assays and efficacy of the compounds, also lines 139-154 in the Materials and Methods section and lines 189-190 in the Results section, respectively, have been modified. In addition, we adjusted Kd to IC50 values in Supplementary Table 1.

4. At least one of the figures with heat maps and inhibition curves can be put into the supplemental information. Some graphs do not have calculated IC50's---please state why?

Response: We agree that not all heat maps need to be in the main body and have moved Figure 5 to the supplement. We have also added more explanatory figure legends explaining the lack of IC50 values for some curves which could not be accurately calculated due to inactivity in the assay.

5. As far as I can see in the manuscript, no statistical analysis was performed on activity of the compounds...were the curves generated once or in duplicate, triplicate?

Response: For the initial figures, data from independent experiments were pooled for generating curves and IC50 values. We have performed a re-analysis of the data to generate IC50 values in triplicate for MOLM13, DU4475, HDQP1, & IGROV1 cell lines. These data are shown in Supplementary Figures S1, S2, S4, & S6. Individual data points are visible as well as the confidence of the curves. Additionally, we have calculated pIC50 values for these compounds for improved visualization of the structure-activity relationship and the SEM for IC50 generation, shown in Figure 3B, Figure 4B, Supplementary Figure S3B, and Supplementary Figure S5B. KASUMI-1 cell experiments (n=1) are now shown in Supplementary Figure S7. Overall, we feel that the conclusions drawn in the text are well-represented by the data.

6. The paper shows negative results, but these are still relevant. So I recommend publication, but much more details are needed to put this entire concept the authors posit into a proper therapeutic context.

Response: Thank you for your recommendation. We hope that the changes to the introduction, results, and discussion are now sufficiently detailed to allow for publication.

Manuscript ID: PONE-D-24-53115

Title: "Development and comparison of single FLT3-inhibitors to dual FLT3/TAF1 inhibitors as an anti-leukemic approach"

Reviewer: 2

Comments:

The authors did in vitro analysis of single and dual FLT3/TAF1 inhibitors for acute myeloid leukemia (AML). They tested several 3i-compounds and provide data on their inhibitory effects on FLT3 kinase and TAF1 bromodomains using cell lines. They also found that there is limited added benefit of TAF1(2) bromodomain inhibition in conjunction with high affinity TK inhibition.

Specific comments:

1. The manuscript concludes that TAF1 bromodomain inhibition does not significantly enhance anti-leukemic effects when combined with FLT3 inhibition. However, it could be more thoroughly discussed, especially across different cell lines, and from other literature that might include more mechanistic analysis on the TAF1 inhibition? Because there is lack of mechanistic studies in this paper, I wonder if authors may have any hypothesis of genetic or proteomic explanations why TAF1 targeting offers minimal benefit?

Response: We agree that a more thorough discussion of TAF1 biology specifically could help the reader. We have added additional text to both the introduction and discussion on evidence for the role of TAF1 in cancer and specific mechanisms that could explain both the potential for therapeutic targeting and the lack of activity in the present study (line 73-78).

“There is also genetic evidence for the role of TAF1 in cancer, as mutations in TAF1 were implicated as drivers of clear cell endometrial cancer [31], and knockdown of TAF1 resulted in decreased proliferation and self-renewal in leukemia cells expressing a splice variant of the AML1-ETO fusion protein [32]. Additionally, TAF1 was shown to interact with acetylated p53 via its bromodomain to mediate transcriptional activation and promote p53 degradation via phosphorylation [33,34]. These studies imply that TAF1 inhibition might be a suitable therapeutic target in a broad set of cancers.”

2. The main conclusion was mainly from cell lines such as MOLM13 and KASUMI-1, and I'm not sure if that is sufficient to draw such conclusions, might need to discuss that, especially if further analysis (with primary cells/in vivo) will be needed to confirm the finding.

Response: We agree that the present experiments are not extendable to all cell lines, and that there might indeed be a cancer cell line (primary or otherwise) that would show added benefit with dual FLT3/TAF1 inhibition compared to single FLT3 inhibition. Indeed, our results in HDQP1 cells (Supplementary Figure S3) suggest this may be the case for some cell lines. We have added text to the discussion (lines 374-383) regarding this key point.

“Admittedly, only a few cell lines were tested in the present study, and it is conceivable that dual FLT3/TAF1 inhibitors might display improved activity over single FLT3 inhibitors at therapeutically relevant concentrations in cell lines with specific genetic mutations. Additionally, different combinations of TK/bromodomain inhibitors might more effectively induce anti-proliferative effects. Further insight into the requirement of specific bromodomain-containing proteins and the bromodomain itself in cancer subtypes would greatly enhance efforts to develop new therapies targeting the transcriptional machinery. However, based on the results of the present study, we concluded that added benefit of TAF1(2) bromodomain inhibition is minimal for the development of anti-leukemic compounds and further focused on optimizing compound activity against tyrosine kinases.”

3. The statistical analysis is unclear, and also the sample size, confidence intervals or p-values for the IC50 differences should be included to substantiate claims.

Response: As mentioned also to the first reviewer, for the initial figures, data from independent experiments were pooled for generating curves and IC50 values. We have performed a re-analysis of the data to generate IC50 values in triplicate for MOLM13, DU4475, HDQP1, & IGROV1 cell lines. These data are shown in Supplementary Figures S1, S2, S4, & S6. Individual data points are visible as well as the confidence of the curves. Additionally, we have calculated pIC50 values for these compounds for improved visualization of the structure-activity relationship and the SEM for IC50 generation, shown in Figure 3B, Figure 4B, Supplementary Figure S3B, and Supplementary Figure S5B. KASUMI-1 cell experiments (n=1) are now shown in Supplementary Figure S7.

4. In fig. 1 legends, could author also describe how they obtain the IC50 numbers?

Response: As stated by the Reviewer, this information has been added into the Figure legend 1 of the revised manuscript on lines 201-205.

“Novel 3i-compound series demonstrates an excellent half maximal inhibitory concentration (IC50) and selectivity for FLT3- and TAF1(2)-inhibition. An 11-point, 3-fold serial dilution of each 3i-compound was assessed for IC50 activity on FLT3 kinase and TAF1(2) bromodomain using assays conducted by Eurofins/DiscoverX in San Diego, USA. B Reference structures and literature-reported IC50 values for gilteritinib (ASP2215), quizartinib (AC220), BAY299 and GNE-371 [20,24,43,44].”

---

## [Decision Letter · Decision Letter 1]

19 Feb 2025

Development and comparison of single FLT3-inhibitors to dual FLT3/TAF1 inhibitors as an anti-leukemic approach

PONE-D-24-53115R1

Dear Dr. Välimäki,

We’re pleased to inform you that your manuscript has been judged scientifically suitable for publication and will be formally accepted for publication once it meets all outstanding technical requirements.

Kind regards,

Francesco Bertolini, MD, PhD

Academic Editor

PLOS ONE

Additional Editor Comments (optional):

Reviewers' comments:

Reviewer's Responses to Questions

**Comments to the Author**

1. If the authors have adequately addressed your comments raised in a previous round of review and you feel that this manuscript is now acceptable for publication, you may indicate that here to bypass the “Comments to the Author” section, enter your conflict of interest statement in the “Confidential to Editor” section, and submit your "Accept" recommendation.

Reviewer #1: All comments have been addressed

Reviewer #2: All comments have been addressed

2. Is the manuscript technically sound, and do the data support the conclusions?

Reviewer #1: Yes

Reviewer #2: Yes

3. Has the statistical analysis been performed appropriately and rigorously? 

Reviewer #1: Yes

Reviewer #2: Yes

4. Have the authors made all data underlying the findings in their manuscript fully available?

Reviewer #1: Yes

Reviewer #2: Yes

5. Is the manuscript presented in an intelligible fashion and written in standard English?

Reviewer #1: Yes

Reviewer #2: Yes

6. Review Comments to the Author

Reviewer #1: all comments addressed adequately manuscript is in proper english syntax and usage--statistics were included

Reviewer #2: The authors have addressed my previous questions and comments, thus I recommend the publication of this manuscript in PlosOne

7. PLOS authors have the option to publish the peer review history of their article (what does this mean? ). If published, this will include your full peer review and any attached files.

**Do you want your identity to be public for this peer review?** For information about this choice, including consent withdrawal, please see our Privacy Policy .

Reviewer #1: No

Reviewer #2: No

---

## [Editor Report · Acceptance letter]

PONE-D-24-53115R1

PLOS ONE

Dear Dr. Välimäki,

I'm pleased to inform you that your manuscript has been deemed suitable for publication in PLOS ONE. Congratulations! Your manuscript is now being handed over to our production team.

Kind regards,

on behalf of

Dr. Francesco Bertolini

Academic Editor

PLOS ONE